# Addressing challenges with real-world synthetic control arms to demonstrate the comparative effectiveness of Pralsetinib in non-small cell lung cancer

Sanjay Popat [1], Stephen V. Liu[2], Nicolas Scheuer[3], Grace G. Hsu [4], Alexandre Lockhart[4], Sreeram V. Ramagopalan [5✉], Frank Griesinger[6] & Vivek Subbiah [7✉]

As advanced non-small cell lung cancer (aNSCLC) is being increasingly divided into rare oncogene-driven subsets, conducting randomised trials becomes challenging. Using real-world data (RWD) to construct control arms for single-arm trials provides an option for comparative data. However, non-randomised treatment comparisons have the potential to be biased and cause concern for decision-makers. Using the example of pralsetinib from a RET fusion-positive aNSCLC single-arm trial (NCT03037385), we demonstrate a relative survival benefit when compared to pembrolizumab monotherapy and pembrolizumab with chemotherapy RWD cohorts. Quantitative bias analyses show that results for the RWD-trial comparisons are robust to data missingness, potential poorer outcomes in RWD and residual confounding. Overall, the study provides evidence in favour of pralsetinib as a first-line treatment for RET fusion-positive aNSCLC. The quantification of potential bias performed in this study can be used as a template for future studies of this nature.

---

[1] Royal Marsden Hospital and Institute of Cancer Research, London, UK. [2] Lombardi Comprehensive Cancer Center, Georgetown University, Washington, DC, USA. [3] Roche Products Ltd, Welwyn Garden City, UK. [4] Cytel, Inc, Waltham, MA, USA. [5] Global Access, F. Hoffmann-La Roche, Basel, Switzerland. [6] Department of Medical Oncology, Pius-Hospital Oldenburg, Oldenburg, Germany. [7] The University of Texas MD Anderson Cancer Center, Houston, TX, USA. ✉email: sreeram.ramagopalan@roche.com; vsubbiah@mdanderson.org

The advent of immune checkpoint inhibitors and molecularly targeted therapy have altered the landscape of non-small cell lung cancer[1] (NSCLC). Randomised trials have shown the benefit of targeted therapy in EGFR- and ALK- driven NSCLC over standard-of-care immunotherapy and chemo-immunotherapy. As NSCLC is being increasingly divided into rare oncogene-driven subsets, it is becoming challenging and infeasible to conduct well-powered randomised trials. In some cases, there is a lack of clinical equipoise when randomising to standard traditional therapies once a rationally designed targeted therapy produces high response rates with impressive durability in single-arm studies.

ARROW is a multi-cohort, open-label, phase I/II study (NCT03037385) that demonstrated that pralsetinib, a highly potent selective RET inhibitor, was efficacious when administered to treatment-naïve patients with advanced RET fusion-positive NSCLC[2,3]. Given the promising results of pralsetinib demonstrated in ARROW, the comparative effectiveness of pralsetinib relative to other therapies amongst patients with advanced NSCLC (aNSCLC) in terms of time-to-treatment discontinuation (TTD), overall survival (OS) and progression-free survival (PFS) is currently unknown and of interest. Whilst a front-line randomized phase III trial is ongoing, the feasibility of a definitive outcome from this strategy for a rare-molecular subset remains uncertain due to recruitment challenges with an efficacious intervention on a background of significant COVID infections. To fill this evidence gap, one of the two goals of this study was to investigate the relative effectiveness of pralsetinib by comparing outcomes for RET fusion-positive patients receiving first-line (1 L) pralsetinib in the ARROW trial with synthetic control arms (SCAs) derived from real-world data (RWD).

Drawing upon RWD to construct SCAs for comparison is used for situations where running a randomised clinical trial (RCT) is impractical or infeasible, or where RCT data is currently unavailable. However, non-randomised treatment comparisons have the potential to be biased due to unmeasured confounding, missing data in RWD and potential poorer performance of an RWD SCA as compared to the pivotal trial the comparator medicine was approved on. These issues have all caused concern for decision-makers evaluating SCA comparisons[4] and suggestions have been made by both regulators and health technology assessment (HTA) agencies that the validity of any conclusions should be supported by analyses that quantify the impact of potential sources of bias[5–7]. Thus, the second of the two goals of this study was to demonstrate how we quantitatively assessed the robustness of our findings to potential sources of bias in a comprehensive and systematic fashion to act as a guide for future SCA studies using RWD[8,9].

With the first goal to investigate the comparative effectiveness of pralsetinib relative to other therapies, we had to consider that since the prevalence of RET fusions is low in NSCLC (1–2%)[10], using RWD for studies involving patients with RET fusion-positive status would be challenging. Thus, since we expected to have a limited number of RET fusion-positive patients available in RWD sources, and the prognostic value of RET fusion status appears to be limited based on the evidence currently available[11–14], additional comparisons with aNSCLC patients of RET fusion unknown status were assessed. This allowed us to maximise the sample sizes of the RWD cohorts, translating into much higher statistical power and ability to adjust for imbalances in patient characteristics between cohorts.

## Results

**Demographics and clinical characteristics**. The CGDB RET fusion-positive 1 L best-available therapy (BAT) (definition in

**Table 1 Baseline characteristics of the 1 L ARROW trial participants given pralsetinib and 1 L CGDB cohort given the best available therapy without adjustment.**

| Baseline characteristics | Best available therapy | Pralsetinib | SMD |
|---|---|---|---|
| Sample size – n | 10 | 116 | – |
| Age ≥ 65 – n (%) | 5 (50.0) | 49 (42.2) | 0.156 |
| Male – n (%) | 2 (20.0) | 55 (47.4) | 0.606 |
| Stage IV – n (%) | 7 (70.0) | 95 (81.9) | 0.281 |
| Smoking status – n (%) | | | |
| History of smoking | 4 (40.0) | 45 (38.8) | 0.23 |
| No history of smoking | 6 (60.0) | 68 (58.6) | |
| Unknown | 0 (0.0) | 3 (2.6) | |
| ECOG – n (%) | | | |
| 0 | 5 (50.0) | 35 (30.2) | 1.017 |
| 1 | 3 (30.0) | 80 (69.0) | |
| 2 | 0 (0.0) | 1 (0.9) | |
| Missing | 2 (20.0) | 0 (0.0) | |
| Non-squamous histology – n (%) | 10 (100.0) | 115 (99.3) | 0.132 |
| Time since diagnosis – median (IQR) | 1.50 (0.77, 8.77) | 1.76 (1.25, 2.51) | 0.069 |
| Metastases: Brain/CNS site – n (%) | 1 (10.0) | 31 (26.7) | 0.442 |
| Race – n (%) | | | |
| Other | 2 (20.0) | 53 (45.7) | 0.664 |
| Unknown | 2 (20.0) | 6 (5.2) | |
| White | 6 (60.0) | 57 (49.1) | |

the Supplementary – 1 L BAT regimens) had 10 patients in total, and baseline patient characteristics are shown in Table 1. Notable imbalances between the pralsetinib and Clinico-Genomic Database (CGDB)[15] cohorts were observed for sex, Eastern Cooperative Oncology Group (ECOG) Performance Status (PS) score, and race (SMD [standardized mean difference] >0.6).

For the comparison between the pralsetinib and enhanced data-mart (EDM) pembrolizumab cohorts in 1 L, there were 795 patients in total, and the comparison with pralsetinib and EDM pembrolizumab with chemotherapy cohorts in 1 L had 1379 patients in total: 109 in the pralsetinib trial cohort, 686 in the pembrolizumab EDM cohort, and 1270 in the pembrolizumab with chemotherapy EDM cohort. Clinical and demographic characteristics of patients are shown in Table 2. As expected, there were more smokers in the RWD cohort for both comparisons relative to the pralsetinib cohort (Table 2).

### Comparative effectiveness

*CGDB RET fusion-positive comparison*. Given sample size, an unadjusted comparison was performed between ARROW and the CGBD cohort. The unadjusted comparison between the pralsetinib and CGDB RET fusion-positive 1 L BAT cohorts showed that pralsetinib was associated with higher TTD, OS, and PFS. The hazard ratios (HRs) were TTD 0.71 (95% CI [confidence interval], 0.34–1.48), OS 0.45 (95% CI, 0.16–1.25), and PFS 0.71 (95% CI, 0.32–1.55); these associations however were limited by sample size (N = 10).

*Pembrolizumab monotherapy EDM comparison*. Following inverse probability of treatment weighting (IPTW)[16], sufficient balance based on a conservative cut-off of SMDM <0.1 was achieved for sex, ECOG PS, time from initial diagnosis, and stage at diagnosis for the comparison between the pralsetinib and pembrolizumab cohorts (Table 3). Age, smoking history, and race demonstrated residual imbalance, though all have SMD <0.25, which has also been suggested as a reasonable threshold for

**Table 2 Baseline characteristics of the 1 L ARROW trial participants given pralsetinib and Flatiron EDM cohort given 1 L pembrolizumab, and 1 L pembrolizumab with chemotherapy without adjustment; balanced variables are those with SMD < 0.1.**

| | Level | Pembrolizumab | Pralsetinib | SMD | Pembrolizumab with chemotherapy | Pralsetinib | SMD |
|---|---|---|---|---|---|---|---|
| N | | 686 | 109 | | 1270 | 109 | |
| Age (%) | <65 | 197 (28.7) | 65 (59.6) | 0.655 | 508 (40.0) | 65 (59.6) | 0.4 |
| | >=65 | 489 (71.3) | 44 (40.4) | | 762 (60.0) | 44 (40.4) | |
| Sex (%) | F | 375 (54.7) | 59 (54.1) | 0.011 | 569 (44.8) | 59 (54.1) | 0.187 |
| | M | 311 (45.3) | 50 (45.9) | | 701 (55.2) | 50 (45.9) | |
| Smoking history at baseline (%) | History of smoking | 628 (91.5) | 43 (39.4) | 1.31 | 1144 (90.1) | 43 (39.4) | 1.25 |
| | No history of smoking | 58 (8.5) | 66 (60.6) | | 126 (9.9) | 66 (60.6) | |
| ECOG (%) | 0 | 230 (33.5) | 34 (31.2) | 0.05 | 512 (40.3) | 34 (31.2) | 0.191 |
| | 1 | 456 (66.5) | 75 (68.8) | | 758 (59.7) | 75 (68.8) | |
| Time from initial diagnosis to first dose (months) (median [IQR]) | | 1.41 [0.92, 2.85] | 1.74 [1.25, 2.30] | 0.054 | 1.18 [0.76, 1.84] | 1.74 [1.25, 2.30] | 0.148 |
| Stage at initial diagnosis (%) | STAGE I, II, or III | 192 (28.0) | 17 (15.6) | 0.304 | 204 (16.1) | 17 (15.6) | 0.013 |
| | STAGE IV | 494 (72.0) | 92 (84.4) | | 1066 (83.9) | 92 (84.4) | |
| Race (%) | White | 493 (71.9) | 54 (49.5) | 0.612 | 883 (69.5) | 54 (49.5) | 0.573 |
| | Other | 123 (17.9) | 49 (45.0) | | 248 (19.5) | 49 (45.0) | |
| | Unknown | 70 (10.2) | 6 (5.5) | | 139 (10.9) | 6 (5.5) | |
| Brain/CNS metastasis only (%) | 0 | 597 (87.0) | 79 (72.5) | 0.368 | 1090 (85.8) | 79 (72.5) | 0.333 |
| | 1 | 89 (13.0) | 30 (27.5) | | 180 (14.2) | 30 (27.5) | |

**Table 3 Baseline characteristics of the 1 L ARROW trial participants given pralsetinib and Flatiron EDM cohort given 1 L pembrolizumab, and 1 L pembrolizumab with chemotherapy after IPTW-adjustment; balanced variables are those with SMD < 0.1.**

| | Level | Pembrolizumab | Pralsetinib | SMD | Pembrolizumab with chemotherapy | Pralsetinib | SMD | Adjusted |
|---|---|---|---|---|---|---|---|---|
| ESS/n | | 115/683 | 109/109 | | 217/1270 | 109/109 | | |
| Age (%) | <65 | 48.3 | 59.6 | 0.23 | 58.9 | 59.6 | 0.015 | Y |
| | >=65 | 51.7 | 40.4 | | 41.1 | 40.4 | | |
| Sex (%) | F | 50.6 | 54.1 | 0.072 | 54.5 | 54.1 | 0.007 | Y |
| | M | 49.4 | 45.9 | | 45.5 | 45.9 | | |
| Smoking history at baseline (%) | History of smoking | 48.9 | 39.4 | 0.192 | 40.3 | 39.4 | 0.017 | Y |
| | No history of smoking | 51.1 | 60.6 | | 59.7 | 60.6 | | |
| ECOG (%) | 0 | 27.8 | 31.2 | 0.075 | 32.9 | 31.2 | 0.037 | Y |
| | | 72.2 | 68.8 | | 67.1 | 68.8 | | |
| Time from initial diagnosis to first dose (months) (median [IQR]) | 1 | 1.45 [0.92, 2.45] | 1.74 [1.25, 2.30] | 0.078 | 1.32 [0.92, 2.24] | 1.74 [1.25, 2.30] | 0.042 | Y |
| Stage at initial diagnosis (%) | STAGE I, II, or III | 17 | 15.6 | 0.038 | 16.6 | 15.6 | 0.028 | Y |
| | STAGE IV | 83 | 84.4 | | 83.4 | 84.4 | | |
| Race (%) | White | 56.7 | 49.5 | 0.199 | 52.3 | 49.5 | 0.061 | Y |
| | Other | 35.6 | 45 | | 41.9 | 45 | | |
| | Unknown | 7.7 | 5.5 | | 5.8 | 5.5 | | |
| CNS metastases only (%) | 0 | 82.5 | 72.5 | 0.241 | 87.5 | 72.5 | 0.383 | N |
| | 1 | 17.5 | 27.5 | | 12.5 | 27.5 | | |

*ESS* Effective sample size; the sample size of an unweighted sample which incorporates the precision of the given weighted sample, *n* number of patients in remaining in IPTW-trimmed sample.

balance[17]. The central nervous system (CNS) metastases variable remained imbalanced (SMD = 0.241), but recording of metastases differs between ARROW and the EDM. The ESS of the pembrolizumab group was 115.

For the comparisons between the pralsetinib trial cohort and EDM 1 L pembrolizumab cohort, post-IPTW-adjustment, pralsetinib was associated with significantly higher TTD, OS, and PFS.

The adjusted HRs for the comparison were TTD 0.49 (95% CI, 0.33–0.73), OS 0.33 (95% CI, 0.18–0.61), PFS 0.47 (95% CI, 0.31–0.7),

*Pembrolizumab and chemotherapy EDM comparison.* For the comparison between the pralsetinib and EDM pembrolizumab with chemotherapy groups, following IPTW-adjustment, balance

was achieved for age, smoking history, race, sex, ECOG PS, time from initial diagnosis, and stage at diagnosis based on a threshold of SMD <0.1 (Table 3). Indeed, only CNS metastases, appeared to have residual imbalance. The ESS of the pembrolizumab and chemotherapy group was 217.

For the comparisons between the pralsetinib trial cohort and EDM 1 L pembrolizumab with chemotherapy cohort, post-IPTW-adjustment, pralsetinib was associated with significantly higher TTD, OS, and PFS. The adjusted HRs for the were TTD 0.5 (95% CI, 0.36–0.7), OS 0.36 (95% CI, 0.21–0.64), PFS 0.5 (95% CI, 0.36–0.7) as shown in Fig. 1.

Sensitivity analyses corresponding to comparisons with the CGDB RET fusion-positive 1 L BAT cohort were not executed due to sample size considerations. Thus, we present the key results from the comparisons with the EDM cohorts in the following sections.

**Sensitivity analysis – Quantitative Bias Analysis (QBA) for missing data assumptions about baseline covariates.** In the pembrolizumab cohort, ECOG PS was missing for 294 patients (30%), and in the pembrolizumab with chemotherapy cohort, for 449 patients (26%). Following multiple imputation of ECOG PS scores, the 1 L comparison between pralsetinib and EDM pembrolizumab and pembrolizumab with chemotherapy, pralsetinib was still associated with significantly higher OS and the adjusted HRs were 0.38 (95% CI 0.21–0.67) and 0.37 (95% CI 0.21–0.64) respectively.

Tipping point-based bias analysis assuming non-random missingness for ECOG PS was executed. As no tipping points could be identified for either comparison of pralsetinib with pembrolizumab or pembrolizumab with chemotherapy for OS, this indicated that the adjusted HRs are robust to extreme deviations from random missingness for baseline ECOG PS. The MAR (data missing at random) and MNAR (data missing not at random) analyses showed our results were also robust in general to missingness assumptions for measured baseline covariates under standard multiple imputation compared to the main analyses.

**Sensitivity analysis – Impact of metastases.** The EDM pembrolizumab cohort had 365 patients (53.2%) without recorded metastases, and the EDM pembrolizumab with chemotherapy cohort also had a large proportion of 582 patients (45.8%) with no record of metastases. IPTW-based analyses including metastases in the propensity score model still yielded significantly better adjusted HRs in favour of pralsetinib for TTD 0.59 (95% CI, 0.38–0.93), OS 0.29 (95% CI, 0.15–0.57), and PFS 0.45 (95% CI, 0.29–0.71) in comparisons with the pembrolizumab cohort, and significantly better TTD 0.42 (95% CI, 0.30–0.60), OS 0.31 (95% CI 0.17–0.54), and PFS 0.38 (95% CI, 0.26–0.54) in the comparisons using the pembrolizumab with chemotherapy cohort.

**Sensitivity analysis – QBA of unmeasured confounding.** In Fig. 2, we plotted bias curves for 1 L pralsetinib vs EDM 1 L pembrolizumab and 1 L pralsetinib vs EDM 1 L pembrolizumab with chemotherapy comparisons. The black curve at the point estimate of 0.38 (95% CI 0.21–0.67; ARR 0.51) in Fig. 2A plots the range of values for the association of a confounder with survival and treatment assignment that would be needed to nullify our conclusions, i.e., that the resulting unconfounded effect estimate would equal 1 on the risk ratio (RR) scale for the pralsetinib versus pembrolizumab comparison. In Fig. 2B, for the comparison between pralsetinib and pembrolizumab with chemotherapy,

the black curve was plotted at the point estimate 0.37 (95% CI 0.21–0.67).

The E-value on the RR scale, was 3.31 for the comparison of 1 L pralsetinib with EDM 1 L pembrolizumab, and 3.37 for the comparison with EDM 1 L pembrolizumab and chemotherapy. Amongst measured covariates, the highest association with the outcome OS was observed for age, and the highest association with exposure was smoking history. Therefore, consistent with the bias plots, we expect our results are robust to plausible unmeasured confounding since the QBA suggested it would be implausible for sufficiently large systematic differences in unmeasured prognostic variables to reverse our findings.

**Sensitivity analysis – QBA of hazard ratio robustness for poorer RWD performance.** For the comparison between 1 L pralsetinib and EDM 1 L pembrolizumab cohorts, at the transformation threshold, the EDM OS curve is well above that of KEYNOTE-42 (Fig. 3A), which has a median OS of 16.7 months (95% CI 13.9–19.7)[18]. The median OS of the untransformed true EDM cohort was 19.17 months (95% CI 10.22-NA) with an IPTW-adjusted HR of 0.35 (95% CI 0.19-0.64), and at the transformation threshold, the median OS was 32.58 months (95% CI 17.38-NA), with an IPTW-adjusted HR of 0.53 (95% CI 0.29–0.96).

For the comparison between 1 L pralsetinib and EDM 1 L pembrolizumab with chemotherapy cohorts, at the transformation threshold, the EDM OS curve is above that of KEYNOTE-189[19] (Fig. 3B), which has a median OS of 22.0 months (95% CI 19.5–25.2). The median OS of the untransformed EDM cohort was 15.75 months (95% CI 12.46-31.36) with an IPTW-adjusted HR of 0.37 (95% CI 0.21–0.65), and at the transformation threshold, the median OS was 25.20 months (95% CI 19.94-NA), with an IPTW-adjusted HR of 0.56 (95% CI 0.32-0.99).

**Discussion**
This study directly compares OS, PFS, and TTD outcomes for pralsetinib versus other first-line treatments in the real-world for aNSCLC. The two goals of this study were to investigate the effectiveness of pralsetinib by constructing an SCA for the ARROW study from RWD, and secondly demonstrate the application of multiple QBA methods to quantify a number of potential sources of bias. This demonstration is motivated by the current landscape where even though propensity-score based methods are commonly used for indirect comparisons and can mitigate the effects of selection bias, many of these studies do not seek to quantify the effects of other types of bias, making it difficult to assess the robustness of resulting estimates, as highlighted by regulators and HTA agencies[20–27].

Being a rare mutation at 1%-2% of NSCLC[10] along with limited testing uptake over time, we expected that there would be a prohibitive number of RET fusion-positive patients available in RWD sources. Thus, this study involved comparisons between RET fusion-positive patients from the ARROW trial to two types of RWD patient groups: 1) the subset of RET fusion-positive patients from the CGDB, and 2) RET fusion status unknown patients from the EDM, which has many more patients than the CGDB. The assumption based on currently available evidence that RET fusion status is not distinctly prognostic allowed for flexibility in using the EDM for cohort development[11–14].

The comparisons involving the CGDB RET fusion-positive 1 L BAT and 1 L pralsetinib cohorts showed that pralsetinib was associated with higher TTD, OS, and PFS, though these associations however were limited by sample size (N = 10). The comparisons using cohorts drawn from the EDM showed significant association in favour of pralsetinib, as well as far greater

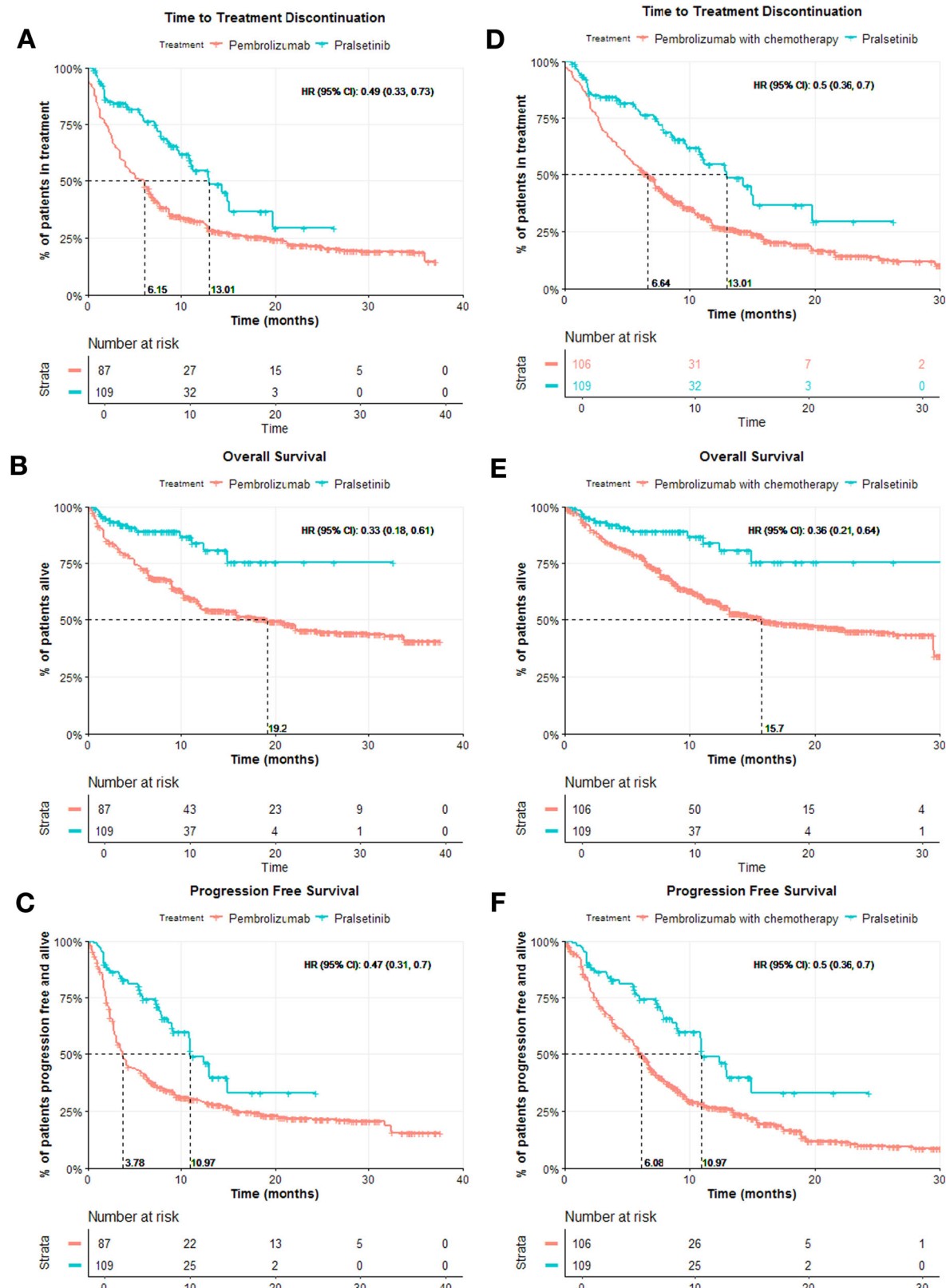

**Fig. 1 Kaplan–Meier curves for TTD, OS, and PFS for comparisons between the 1 L pralsetinib trial cohort and 1 L pembrolizumab cohort, and 1 L pralsetinib trial cohort and 1 L pembrolizumab with chemotherapy cohort. A–C** The Kaplan–Meier curves are for each endpoint TTD, OS, and PFS panels respectively for the comparison between 1 L pralsetinib versus 1 L pembrolizumab (ESS = 109 for the pralsetinib cohort, and ESS = 115 for the pembrolizumab cohort), and **D–F** for the comparison between 1 L pralsetinib versus 1 L pembrolizumab with chemotherapy after IPTW-adjustment (ESS = 109 for the pralsetinib cohort, and ESS = 217 for the pembrolizumab with chemotherapy cohort); the median OS for the pralsetinib cohorts could not be computed.

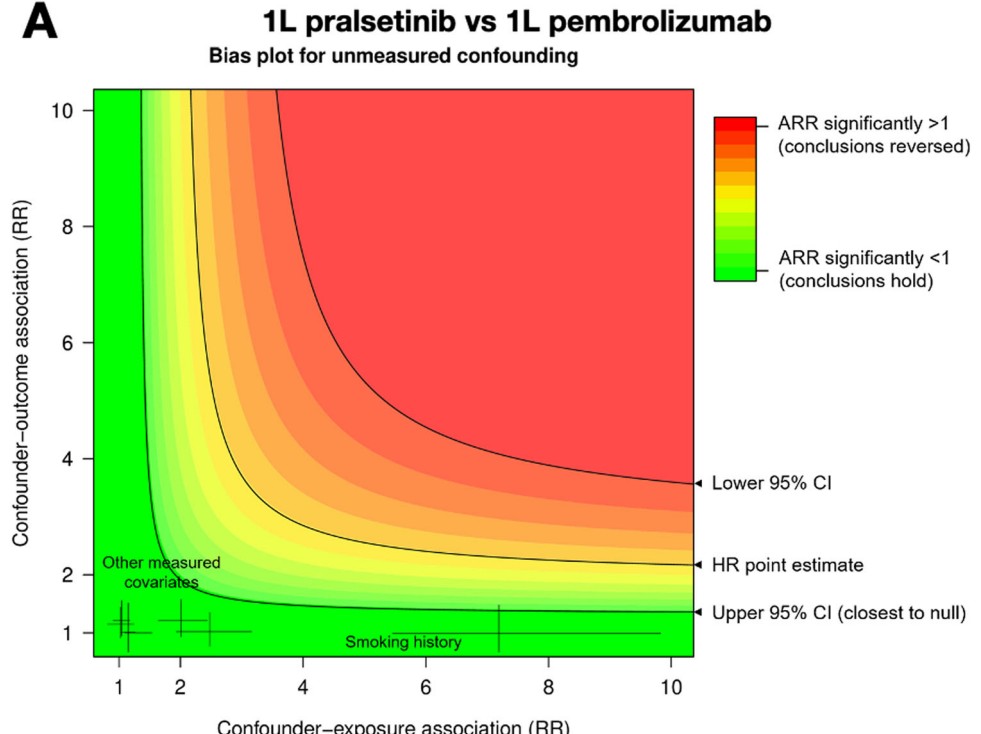

**Fig. 2 Bias plots showing unmeasured confounding for comparisons between the 1L pralsetinib trial cohort and 1L pembrolizumab cohort, and 1L pralsetinib trial cohort and 1L pembrolizumab with chemotherapy cohort. A** Bias plots for unmeasured confounding corresponds to the comparison with 1L pembrolizumab comparison (HR 0.38, 95% CI 0.21–0.67), **B** corresponds to the comparison involving 1L pembrolizumab with chemotherapy comparison (HR 0.37, 95% CI 0.21–0.64). These graphs plot unconfounded treatment effect estimates as risk ratios (ARR adjusted risk ratio) after adjusting for a hypothetical unmeasured binary confounder over a range of confounder-exposure and confounder-outcome associations on the risk ratio scale. The colors map the strength of an unmeasured confounder (x and y axes) to the robustness of this study's conclusions (color gradient). The worst-case strengths of measured baseline confounders are shown using HRs from the multiple imputation resulting from QBA for missing data assumptions.

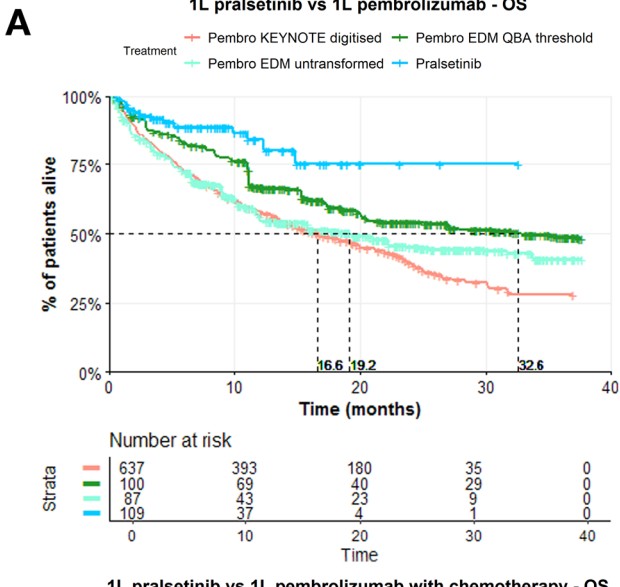

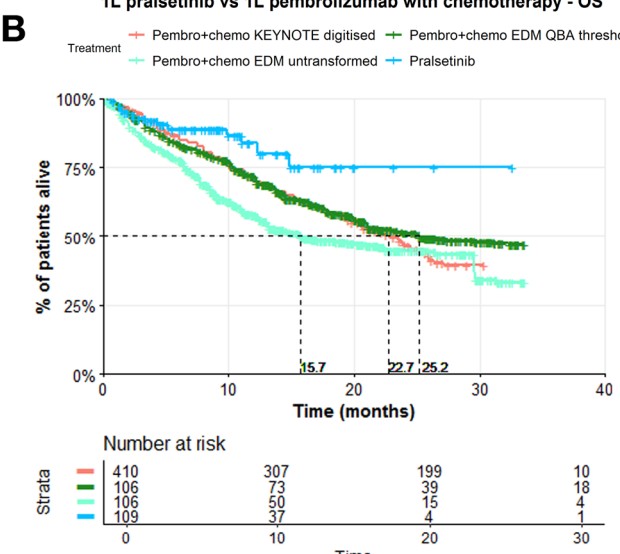

**Fig. 3 Kaplan–Meier curves for OS showing the robustness of the estimated hazard ratios for the comparison between 1 L pralsetinib trial cohort and 1 L pembrolizumab cohort, and 1 L pralsetinib trial cohort and 1 L pembrolizumab with chemotherapy cohort when the RWD cohorts are transformed to have increased OS. A** The Kaplan–Meier curves for OS correspond to the comparison between 1 L pralsetinib versus 1 L pembrolizumab (N = 637 for the digitised KEYNOTE cohort, ESS = 109 for the pralsetinib cohort, ESS = 115 for the pembrolizumab cohort when untransformed and at the transformation threshold), and **B** corresponds to the comparison between 1 L pralsetinib versus 1 L pembrolizumab with chemotherapy (N = 410 for the digitised KEYNOTE cohort, ESS = 109 for the pralsetinib cohort, ESS = 217 for the pembrolizumab with chemotherapy cohort when untransformed and at the transformation threshold) after QBA of the HRs; red indicates the digitised curve from the corresponding KEYNOTE trial (KEYNOTE-42 for pembrolizumab and KEYNOTE-189 for pembrolizumab with chemotherapy), light green indicates the untransformed weighted KM curve for the EDM cohort, dark green is the weighted KM curve for the EDM at the transformation threshold where the adjusted HR remains significant, and blue indicates the pralsetinib group's weighted KM curve.

precision of treatment effect estimates. All comparisons between the pralsetinib trial and RWD EDM cohorts showed that pralsetinib was significantly associated with higher TTD, OS, and PFS over pembrolizumab, and pembrolizumab with chemotherapy. The results from the comparison with 1 L pembrolizumab had some residual imbalance in three patient characteristics. Nonetheless, the extent of the imbalance for all of these variables (SMD < 0.25) has been suggested to be reasonable based on a prior study[17]. Additional considerations were the inclusion of imbalanced confounders post-adjustment in the Cox outcome regression models. This was done to account for the residual confounding that could not be addressed purely by IPTW and subsequent weighted analyses. Further, the sensitivity analyses using QBA methods showed that our results from the comparisons with the EDM cohorts for OS were robust against all types of bias tested.

We performed multiple QBA-type sensitivity analyses to alleviate concerns about trial-RWD comparability by quantifying the effects of missing ECOG PS, unmeasured confounding, and reduced survival of patients from RWD relative to that seen in pivotal clinical trials. An advantage of the way we conducted QBA for missing data is that the researcher does not need to know the true mechanism of missingness since the effect on the treatment effects under multiple different missing data assumptions are tested. Tipping point analyses, which were used when working under the assumption that ECOG is MNAR provides a similar advantage. That is, researchers do not need to make assumptions about how the missingness occurs, but rather only consider whether the tipping point, if one exists, is a plausible scenario that may occur.

The QBA of unmeasured confounding also has practical and clear advantages in the context of studies involving RWD, where data limitations are common. Bias plots offer a visual representation of how the method adjusts for a hypothetical unmeasured confounder over a range of confounder-exposure and confounder-outcome associations. This allows for a nuanced assessment of how robust a treatment effect estimate would be against unmeasured confounders.

We also sought to quantify how our conclusion that pralsetinib is associated with significantly better OS would be reversed when we observe that the performance of a treatment in the real-world is worse as compared to that seen in its corresponding pivotal clinical trial. Without requiring any assumptions as to why the discrepancy occurs, the results of the analysis shows how much better the real-world performance needs to be before the association observed is no longer significant. When comparing the survival of this hypothetical group with the corresponding clinical trial via Kaplan–Meier curves and median values, researchers can judge whether their conclusions would be robust against meaningful non-concordance between real-world and clinical trial concordance. To our knowledge, this approach to answer how robust a treatment estimate is against any advantages conveyed to by poorer performance of treatments in real-world and trial settings has not been done previously. Such situations are often observed, with multiple cases in the context of immunotherapy treatments in NSCLC alone[28].

Concerns that metastases are recorded differently between trial and RWD induced the decision to not use these variables for weight estimation since their inclusion may introduce bias. These variables were also not used for judgement as to whether groups being compared were considered balanced. Nevertheless, assessing the robustness of the main analysis estimates when adjusting for metastases resulted in adjusted HRs that support the conclusions from the main comparisons with the EDM cohorts. Possible sources of bias not addressed in this study include the inconsistent characterization of PFS between the ARROW trial

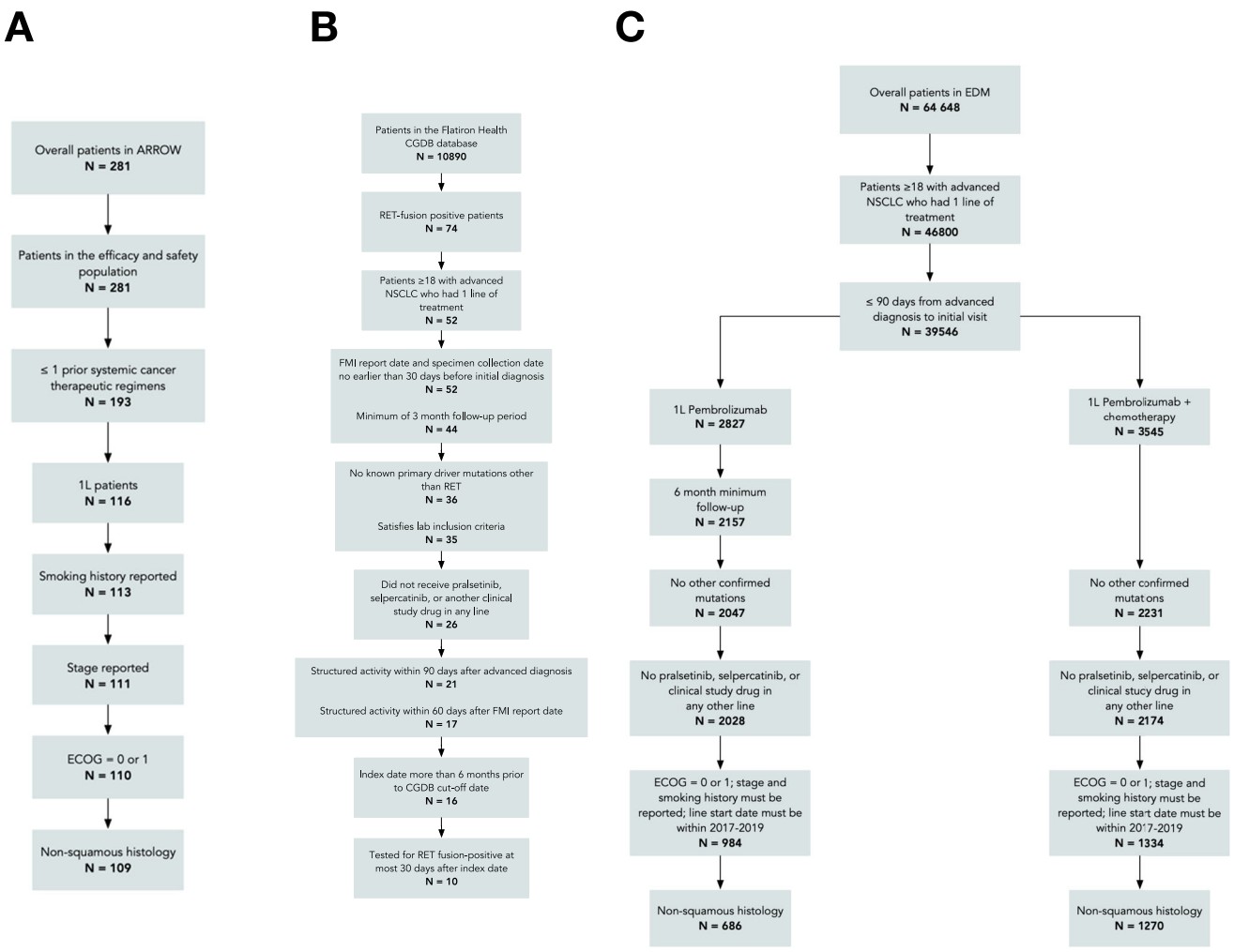

**Fig. 4 Flowcharts describing the patient selection process for the cohorts drawn from the ARROW trial, Flatiron Health CGDB, and Flatiron Health EDM datasets. A** The flowchart corresponds to the ARROW trial, **B** to the Flatiron Health CGDB, and **C** to the Flatiron Health EDM datasets.

and RWD. While we did exclude ALK and EGFR mutations, an additional limitation to be considered for future work is the lack of reporting across the ARROW trial and EDM for other uncommon/uncharacterised oncogene mutations. The effect of these mutations may additionally be exacerbated by differences in smoking status between cohorts.

Overall, this study provides evidence in favour of pralsetinib over pembrolizumab and pembrolizumab with chemotherapy as an effective 1 L treatment for RET fusion-positive aNSCLC. The study also demonstrates multiple sensitivity analyses performed to quantify the effect of multiple sources of bias. In the context of this study, we show that the results of these bias assessments reinforced our findings and can be used as a template for future trial-RWD comparisons.

## Methods

This comparative effectiveness research study adheres to the International Society for Pharmacoeconomics and Outcomes Research (ISPOR) reporting guideline and the Strengthening the Reporting of Observational Studies in Epidemiology (STROBE) reporting guideline for cohort studies[29]. Approval for this study was granted by the WIRB-Copernicus Group institutional review board. Informed consent was waived because the data were deidentified, in accordance with 45 CFR §46.

**Study populations**. The ARROW study (NCT03037385), is a registrational, non-randomized Phase 1 and 2 trial of pralsetinib, in patients with advanced non-resectable NSCLC and other tumours. The trial was conducted at multiple study

sites across the US, Asia, and Europe. The pralsetinib cohort used for the comparisons included patients with RET fusion-positive aNSCLC in the ARROW trial.

The RWD study cohorts were selected from two databases from Flatiron Health. The first of these being the Flatiron Health-Foundation Medicine (FMI) Clinico-Genomic Database (CGDB)[15], a US nationwide, longitudinal database of electronic health records linked to genomic data derived from FMI comprehensive genomic profiling (CGP) tests by deidentified, deterministic matching. An advantage of the CGDB is that it contains test results for RET fusion status. RWD cohorts were also drawn from a second Flatiron Health database, the enhanced data-mart (EDM). The EDM's strength is its large number of patients, though does not have genomic testing information available on patients' RET fusion status[15,30]. Hence, since the prognostic value of RET fusion status appears limited based on the evidence currently available, under the assumption that RET fusion status is not prognostic, the sample size of the RWD cohorts could be maximised by using the EDM[11–14].

The flowcharts in Fig. 4 describe the patient selection process drawn from the (A) ARROW trial, (B) Flatiron Health CGDB, and (C) Flatiron Health EDM cohorts. For the CGDB and EDM study cohorts, patients missing a date of death were censored at their last recorded visit in the database or March 1, 2020 (data cut and study cut-off date), whichever was earlier. The main cohort was from the CGDB: RET fusion-positive receiving 1 L best-available therapy (BAT) (definition in the Supplementary – 1 L BAT regimens). 1 L treatments for this cohort were pooled as the sample size was small. Two other EDM cohorts involved in the head-to-head comparisons where we assumed RET fusion status is not prognostic were selected from the EDM: patients receiving 1 L pembrolizumab, and patients receiving 1 L pembrolizumab with chemotherapy; the chemotherapy was carboplatin and pemetrexed.

Patients from the pralsetinib cohort and the RWD study cohorts had unresectable, locally advanced, or metastatic NSCLC diagnosed between January 1, 2011 and September 1, 2019, had non-squamous histology, and had an Eastern Cooperative Oncology Group (ECOG) Performance Status (PS) score of 0 or 1 at time of 1 L treatment initiation. The following criteria were also applied to the

RWD cohorts: patients do not have EGFR, ALK, ROS1, or BRAF mutations at the date of initiation of 1 L regimen ("index date"), are aged 18 years or older, have <90-day gap between aNSCLC diagnosis and first visit or medication administration, have an index date >6 months prior to the administrative cut-off date of March 1, 2020, a 1 L start date between 2017 and 2019 in order to align with the ARROW trial, and patients could not have pralsetinib or selpercatinib or clinical study drugs in any line of treatment. Identical eligibility criteria were used to select patients for all treatment regimens of interest. The eligibility criteria for the CGDB RET fusion-positive cohort was largely similar to the EDM cohorts (Fig. 4).

Digitised approximations of the Kaplan–Meier curves corresponding to two final phase-3 KEYNOTE trial arms (KEYNOTE-42 and KEYNOTE-189)[21,22] for pembrolizumab monotherapy and pembrolizumab with chemotherapy used for sensitivity analyses. The purpose being to assess for each of the two regimens the impact of any discrepancy in overall survival between the corresponding clinical trial and RWD cohorts.

**Statistical analysis – Comparative effectiveness**. Inverse probability of treatment weighting (IPTW)[16] was used to adjust for differences in patient characteristics between the ARROW trial and RWD cohorts. Estimating the relative treatment effect for a population of patients with similar characteristics to patients from the ARROW trial was of interest. Thus, the chosen estimand was the average treatment effect among the treated (ATT). Unadjusted and IPTW-adjusted hazard ratios (HR) for time to treatment discontinuation (TTD), overall survival (OS) and progression-free survival (PFS), were estimated using Cox proportional-hazards models. Covariates with residual imbalance after IPTW (standardized mean difference [SMD] > 0.1)[17] were controlled by including them as covariates in the Cox model. Missing data was assumed to be completely missing at random and the significance level was set at 5% for all analyses. Unadjusted and IPTW-adjusted Kaplan–Meier (KM) curves were used to estimate median values of TTD, OS, PFS. When IPTW was used, the 95% confidence intervals (CI) were derived using a robust variance estimator. The proportional hazards assumption was justified for all models based on the Schoenfeld test, examination of KM plots and log-negative-log (LNL) plots. The effective sample size (ESS)[31] was used to represent sample size post-IPTW.

**Statistical analysis – Sensitivity analyses**
*Quantitative Bias Analysis (QBA) for missing data assumptions about baseline covariates*. To assess the sensitivity of our results to missing data assumptions, hazard ratios (HRs) were computed under three scenarios:

1. Baseline confounder data missing completely at random (MCAR); these correspond to the results from the primary analysis
2. Baseline confounder data missing at random (MAR)
3. ECOG PS missing not at random (MNAR)

Multiple imputation by chained equations of ECOG PS scores was performed under MAR and MNAR, then where consistent with eligibility criteria for this study, patients with imputed ECOG PS > 1 were excluded, and the comparisons of interest executed[8].

Tipping point bias analysis is an approach to manipulate scenarios for missingness or unmeasured confounding needed to evaluate the robustness of study results. The merits of evaluating the sensitivity of treatment effects under these scenarios is relevant to the common nature of uncollected or missing data in real-world studies. Therefore, it is essential to establish relevant thresholds for common sources of bias in real-world data and better specify the conditions where treatment effect conclusions may hold. For this study, tipping point-based bias analysis assuming non-random missingness (MNAR) for ECOG PS was also used, which involved shifting the distribution of imputed baseline ECOG PS within the RWD groups to poorer than expected under MAR to assess whether the corresponding adjusted HRs remained significant or not.

*Impact of adjusting for metastases*. Metastases were not explicitly adjusted for in the analyses as they are known to be under-recorded in the EDM. To evaluate the effect of metastases on the comparisons between the ARROW trial and EDM RWD cohorts, a sensitivity analysis was performed including a categorical metastases variable in the propensity score model.

*QBA of unmeasured confounding*. This analysis was used to assess the robustness of the study by estimating the E-value[8,9,32]. The E-value represents the minimum association of a hypothetical unmeasured confounder with treatment assignment and outcome of interest (OS) on the risk ratio (RR) scale to nullify our estimated HRs. HRs were converted to approximate RRs using a square-root transformation[33]. Bias plots graph unconfounded treatment effect estimates as fully adjusted risk ratios (ARR) after adjusting for a hypothetical unmeasured binary confounder over a range of confounder-exposure and confounder-outcome associations on the RR scale. Technical details are available in the Supplementary.

*QBA of hazard ratio robustness*. In order to quantitatively assess whether the adjusted HR estimates for the comparisons are robust against systematically poorer

OS in RWD as compared to pivotal trials, we used a tipping point analysis to assess how far the OS in the RWD arms can be improved using a multiplicative constant before the IPTW-adjusted HR value loses statistical significance—we call this the "transformation threshold". To maintain a fixed maximum follow-up time, patients were censored if their transformed time to event was greater than the maximum follow-up time in the original data for the reference/untransformed group.

The analyses and figures were performed in R statistical software version 3.3.6 (R Project for Statistical Computing). Further details on techniques are found in the Supplementary – Supplementary methods.

**Reporting summary**. Further information on research design is available in the Nature Research Reporting Summary linked to this article.

## Data availability

The Flatiron Health data used in this study were licensed from Flatiron Health https://flatiron.com/real-world-evidence/. The databases used were the Clinico-Genomic Database (CGDB) and the enhanced data-mart (EDM). These deidentified data may be made available upon request; interested researchers can contact DataAccess@flatiron.com. The clinical data from the ARROW trial were not generated for the purpose of this study. Researchers may request access to individual patient data from the ARROW trial through Roche's data sharing platforms in accordance with the Global Policy on Sharing of Clinical Study Information: http://www.roche.com/research_and_development/who_we_are_how_we_work/clinical_trials/our_commitment_to_data_sharing.htm. Since at the time of publication the ARROW trial is ongoing and covering multiple indications, the study data will be accessible at https://vivli.org/ when the trial is completed for all indications (expected to be in 2024). In the meantime, requests to access individual patient data from the non-small cell lung cancer arm of the ARROW trial described in the current manuscript can be submitted through: https://vivli.org/members/enquiries-about-studies-not-listed-on-the-vivli-platform/ The remaining data are available within the Article and Supplementary Information.

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

## Acknowledgements

This study was funded by F. Hoffmann-La Roche. The funder of the study was involved in the writing of the manuscript and the decision to submit it for publication.

## Author contributions

S.P.- Conceptualization, Formal Analysis, Writing - review & editing. S.V.R. - Conceptualization, Data Curation, Formal Analysis, Funding Acquisition, Methodology, Project Administration, Resources, Supervision, Validation, Visualization, Writing - original draft, Writing - review & editing. A.L. - Conceptualization, Data Curation, Formal Analysis, Methodology, Validation, Visualization, Writing - original draft, Writing - review & editing. G.H. - Conceptualization, Project Administration, Resources, Supervision, Formal Analysis Validation, Visualization, Writing - original draft, Writing - review & editing. S.V.L. - Conceptualization, Formal Analysis, Writing - review & editing. N.S. - Conceptualization, Formal Analysis, Writing - review & editing. F.G. - Conceptualization, Formal Analysis, Writing - review & editing. V.S. - Conceptualization, Formal Analysis, Writing - review & editing.

## Competing interests

S,P. receives honoraria from Boehringer Ingelheim, AstraZeneca, Roche, Takeda and Chugai Pharma; and provides consulting or advisory role for Boehringer Ingelheim, AstraZeneca, Roche, Takeda, Novartis, Pfizer, Bristol-Myers Squibb, MSD, Guardant Health, AbbVie and EMD Serono. Dr Liu has received research funding (to institution) from Alkermes, AstraZeneca, Bayer, Blueprint Medicines Corporation, Bristol-Myers Squibb, Corvus, Debiopharm, Elevation Oncology, Genentech, Lilly, Lycera, Merck, Merus, Pfizer, Rain Therapeutics, RAPT, and Turning Point Therapeutics; has served as consultant or advisory board member to Amgen, AstraZeneca, BeiGene, Blueprint Medicines Corporation, BMS, Catalyst, Daiichi Sankyo, G1 Therapeutics, Genentech/ Roche, Guardant Health, Inivata, Janssen, Jazz Pharmaceuticals, Lilly, Merck/MSD, PharmaMar, Pfizer, Regeneron, and Takeda. Mr Scheuer reported receiving personal fees from Roche, receiving shares from Roche as an employee during the conduct of the study, and reported being an employee of and receiving shares from Novartis outside the submitted work. G.G.H. and A.L. reported receiving funding from Roche during the conduct of the study. S.V.R. reported receiving personal fees from Roche during the conduct of the study. F.G. has consulted or provided expert opinion for AMGEN, AstraZeneca, Bayer, BMS, Boehringer Ingelheim, Celgene, GSK, Lilly, MSD, Novartis, Pfizer, Roche, Siemens, and Takeda; has received fees from Amgen, AstraZeneca, Bayer, Boehringer Ingelheim, BMS, Celgene, GSK, Lilly, MSD, Novartis, Pfizer, Roche, Siemens, and Takeda; and has received funding for scientific research from Amgen, AstraZeneca, Boehringer Ingelheim, BMS, Celgene, GSK, Lilly, MSD, Novartis, Pfizer, Roche, Siemens, and Takeda. V.S. reports research funding/grant support for clinical trials from AbbVie, Agensys, Alfa-sigma, Altum, Amgen, Bayer, Berg Health, Biotherapeutics, Blueprint Medicines Corporation, Boston Biomedical, Boston Pharmaceuticals, Celgene, D3, Dragonfly Therapeutics, Exelixis, Fujifilm, GSK, Idera Pharma, Incyte, Inhibrx, Loxo Oncology, Medimmune, MultiVir, Nanocarrier, National Comprehensive Cancer Network, NCI-CTEP, Novartis, Northwest Biotherapeutics, Pfizer, PharmaMar, Roche/ Genentech, Takeda, Turning Point Therapeutics, UT MD Anderson Cancer Center, and Vegenics; travel support from ASCO, ESMO, Helsinn, Incyte, Novartis, and PharmaMar; consultancy/advisory board participation for Helsinn, Incyte, Loxo Oncology/Eli Lilly, Medimmune, Novartis, R-Pharma US, QED Pharma, and other relationship with Medscape. V.S. is also an Andrew Sabin Family Foundation Fellow at The University of Texas MD Anderson Cancer Center, acknowledges support of The Jacquelyn A. Brady Fund, is supported by NIH grant R01CA242845. MD Anderson Cancer Center Department of Investigational Cancer Therapeutics is supported by the Cancer Prevention and Research Institute of Texas (RP1100584), the Sheikh Khalifa Bin Zayed Al Nahyan Institute for Personalized Cancer Therapy (1U01 CA180964), NCATS Grant UL1 TR000371 (Center for Clinical and Translational Sciences), and the MD Anderson Cancer Center Support Grant (P30 CA016672).
