## [Peer Review File · Nature Communications]

Reviewers' Comments:

Reviewer #1:

Remarks to the Author:

The authors seek to answer an important question for both lung cancer and other areas of precision medicine. When a treatment has been developed for a small population how do you demonstrate to both clinicians and in particular to funders that the treatment is better than what is already available. Traditionally these have been performed through randomised controlled trials but these may not be practical in the context of a very small population, and ethical if the new therapy has a significantly higher response rates than historical treatments.

This paper uses real world data to try and answer this question in the field of lung cancer harbouring a RET fusion; a rare abnormality where potent new inhibitors have been developed but as yet we do not have randomised controlled trials. Such an attempt is challenging due to the fact that with evolving care most patients in the real world database did not have their RET status known (as either positive or negative) and the demographics of patients with RET driven lung cancer differ from those of the "average" patient with lung cancer in particular relation to smoking status, age and in some series gender. These are known prognostic factors.

They use Inverse probability of treatment weighting to adjust for differences in baseline characteristics between the trial cohort and the real world data cohorts with a number of mechanisms to check for bias; this will require a full statistical review as to the methodology which I'm sure you have arranged.

Issues remain in the results and interpretation of this analysis which are in my view not sufficiently addressed in the discussion and this needs to be done to avoid inappropriate conclusions being made from the authors' work. This is most obvious in the analysis of the pembrolizumab cohort.

Even when they happy that they've achieved reasonable balance by the IPTW there is still a reasonable difference between the cohorts. However of more concern is that in the pembrolizumab cohort there remain significant differences even by their criteria in key prognostic measures which are also thought to be predictive in this setting. For example there is a 10% difference in smoking status. This is not only prognostic but also predictive to immunotherapy response (for example see DOI:<https://doi.org/10.1016/j.eclim.2021.100990>).

Age as well as being known to be a prognostic factor may predict for response to immunotherapy (PMCID: PMC7492214). There are also significant differences in CNS disease and race which may impact on the analysis.

Also not discussed is the potential modifying effect of the underlying genomics, particularly in response to immunotherapy. This will vary between the arrow cohort and the real world cohort and will be exacerbated further by the differences in smoking status. We know even in those oncogenes that are commonly found in non smoking related cancer response to immunotherapy can vary (PMID: 34376553) and this can be impacted on by different mutations within the same gene best described within EGFR (for example see PMID: 31086949)

I am not sure from the manuscript presented the purpose behind the tipping analysis with results demonstrated in Figure 3. They suggest that this analysis is to account for the worst survival in real world cohorts than in clinical trials. This is indeed a well described phenomenon; however I feel this is something that need to be taken into account when interpreting the data generated. I'm not sure provided statistical modelling adding in hazard ratios to find out at what tipping point significance is lost is helpful in interpreting the data. I would want more explanation of the merits of this approach and how it helps in this context.

Lastly I would prefer to see numbers at risk under the K-M plots where appropriate

Reviewer #2:

Remarks to the Author:

General comment: The solution to finding treatment effects in rare populations is ripe for new and creative methods. When found such methods can transform medicine dramatically for patients with rare diseases or rare disease subtypes. The authors have presented their strategy to approach this issue. The idea is one of great interest. This article attempts to present a strategy for identifying the treatment effect of pralsetinib, which is a highly selective RET inhibitor used in an open-label single-arm trial, compared to real world data for best available therapy. The manuscript itself has issues that need to be addressed before it accomplishes what the authors claim. If meaningful revisions can be completed, the strategy could be useful and interesting.

Strengths:

- 1) The authors have compiled meaningful clinical trial data and real-world data, have considered multiple sources of bias when combining real world data with current single arm trial data, and have included a reasonable series of methods to address them.
- 2) The topic is of great interest and timely.
- 3) The methods used are available in existing software so can be reproduced.
- 4) The conclusion is well thought and consistent with the findings.

Weaknesses:

- 1) The abstract, article, and study populations focus on identifying treatment effect of pralsetinib for the rare subtype RET fusion-positive aNSCLC using data from a current single arm trial and 3 real world data sets, 2 of which do not have RET fusion-positive information. These are assumed to be 99% RET fusion negative given the population estimate with an assumption that RET fusion status is not prognostic. A third dataset has RET fusion status. and is used to compare best available therapy. The statistical methods and results section of the manuscript focus almost entirely on the large datasets. In the results, additional information from the KEYNOTE-42 study are introduced without being included in the population description or methods. All results for the rare patients are relegated to the supplementary information, but this finding was the point of this investigation as described from the start. This leaves a disjointed story from start to finish, that is confusing to follow and not in line with the stated purpose of the manuscript.
- 2) The methods described take advantage of existing methods, so this is not novel. This paper applies existing methods in a logical order that accomplishes the desired estimate of treatment effect.
- 3) These methods only work when inclusion in the rare subgroup is not prognostic.

Specific Recommendations:

- 1) If the focus is to be the treatment effect for the rare patient subtype, much of the supplemental material would ideally be moved to the main article and vice versa.
 - a. Supplemental Figures 1 and 2 could be combined into a single figure in the main article and replace most of the text from lines 84-123. This figure should also include the CGDB patients. Reading the text without the figure was challenging and the figure is much more accessible.
 - b. Supplementary material rows 246-277 appears to be the results that match the intention of the manuscript and would be better served in the article itself.
 - c. Article Figure 1 shows the effect of large datasets that do not represent the effect in the rare population. This is an important step in the process but could be in the supplemental materials.
 - d. Article Figure 2 could be moved to supplemental materials.
 - e. An additional supplemental figure could be meaningful if you could use the CGDB data for RET fusion negative patients to confirm general similarity for best available treatment for both negative and positive patients to confirm that your data support that RET fusion status is not prognostic.
- 2) Small inconsistencies or omissions that can be easily addressed.
 - a. The number of acronyms makes reading difficult. I recommend keeping only a small number of universally familiar acronyms throughout.
 - b. Lines 10 and 42. aNSCLC is defined as "advanced NSCLC" in line 10 and as "advanced RET fusion-positive NSCLC" in line 42.
 - c. Line 81. Please define WCG
 - d. Lines 239-240. With only 10 patients in the best available treatment RET fusion-positive cohort, I recommend not mentioning statistical significance.
 - e. Line 292 - the KEYNOTE study shows up here for the first time. A little more justification and

introduction for the reader is needed to justify its use, since it is not mentioned in any methods or patient populations above.

f. Line 371 – While there are SMD <0.1 in the table, none are highlighted in green. This statement and table need to be consistent.

g. Line 372. I recommend defining ESS in the table or as a note below the table. Also, I recommend including the actual N after trimming in addition to the ESS in this table. ESS is traditionally used for determination of power and does not prevent providing the actual N.

h. Kaplan-Meier figures – I recommend adding at risk tables to all such figures.

Responses to reviewer comments

Reviewer #1 (Remarks to the Author):

The authors seek to answer an important question for both lung cancer and other areas of precision medicine. When a treatment has been developed for a small population how do you demonstrate to both clinicians and in particular to funders that the treatment is better than what is already available. Traditionally these have been performed through randomised controlled trials but these may not be practical in the context of a very small population, and ethical if the new therapy has a significantly higher response rates than historical treatments.

This paper uses real world data to try and answer this question in the field of lung cancer harbouring a RET fusion; a rare abnormality where potent new inhibitors have been developed but as yet we do not have randomised controlled trials. Such an attempt is challenging due to the fact that with evolving care most patients in the real world database did not have their RET status known (as either positive or negative) and the demographics of patients with RET driven lung cancer differ from those of the “average” patient with lung cancer in particular relation to smoking status, age and in some series gender. These are known prognostic factors.

They use Inverse probability of treatment weighting to adjust for differences in baseline characteristics between the trial cohort and the real world data cohorts with a number of mechanisms to check for bias; this will require a full statistical review as to the methodology which I'm sure you have arranged.

We thank the reviewer for their overall favorable reception of our paper. We agree with all the comments and have provided a point by point response below.

Comment #1

Issues remain in the results and interpretation of this analysis which are in my view not sufficiently addressed in the discussion and this needs to be done to avoid inappropriate conclusions being made from the authors' work. This is most obvious in the analysis of the pembrolizumab cohort.

Even when they happy that they've achieved reasonable balance by the IPTW there is still a reasonable difference between the cohorts. However of more concern is that in the pembrolizumab cohort there remain significant differences even by their criteria in key prognostic measures which are also thought to be predictive in this setting. For example there is a 10% difference in smoking status. This is not only prognostic but also predictive to immunotherapy response (for example see DOI:<https://doi.org/10.1016/j.eclinm.2021.100990>).

Thank you for these great points. In our demographics section we mention several variables that have greater residual imbalance than our a priori threshold of .1, but follow-up with the comment that $<.25$ based on a simulation study may still considered reasonably sufficient. We have additionally added to the discussion mentioning this imbalance a little more:

The results from the comparison with 1L pembrolizumab had some residual imbalance in three patient characteristics. Nonetheless, the extent of the imbalance for all of these variables (SMD <0.25) has been suggested to be reasonable based on a prior study¹⁷. Additional considerations to deal with biased treatment effects were inclusion of these imbalanced characteristics, or further adjustment for confounding in analyses beyond IPTW-analyses in the Cox model.

Comment #2

Age as well as being known to be a prognostic factor may predict for response to immunotherapy (PMCID: PMC7492214). There are also significant differences in CNS disease and race which may impact on the analysis.

We appreciate your suggestion and agree. In the main analysis and metastases sections we include age and race variables in our inverse probability of treatment (IPTW) weighting analysis (main) and then also include metastases in our sensitivity analysis.

Comment #3

Also not discussed is the potential modifying effect of the underlying genomics, particularly in response to immunotherapy. This will vary between the arrow cohort and the real world cohort and will be exacerbated further by the differences in smoking status. We know even in those oncogenes that are commonly found in non smoking related cancer response to immunotherapy can vary (PMID: 34376553) and this can be impacted on by different mutations within the same gene best described within EGFR

(for example see PMID: 31086949)

We appreciate your comments on the relevance of genomics in our study question. We recognize the relevance of incorporating this into our discussion and posed as an additional limitation in the discussion with:

“While we did exclude ALK and EGFR mutations, an additional limitation to be considered in future studies is the lack of reporting across ARROW and EDM for other treatment modifying oncogene mutations which can be exacerbated by differences in smoking status between cohorts.”

Comment #4

I am not sure from the manuscript presented the purpose behind the tipping analysis with results demonstrated in Figure 3. They suggest that this analysis is to account for the worst survival in real world cohorts than in clinical trials. This is indeed a well described phenomenon; however I feel this is something that need to be taken into account when interpreting the data generated. I'm not sure provided statistical modelling adding in hazard ratios to find out at what tipping point significance is lost is helpful in interpreting the data. I would want more explanation of the merits of this approach and how it helps in this context.

Your comments on the tipping point analysis are well-noted. We have added clarification to the methods pertaining to the merits and how it helped in this context with:

“Tipping point bias analysis is an approach to manipulate scenarios for missingness or unmeasured confounding needed to evaluate the robustness of study results. The merits of evaluating the sensitivity of treatment effects under these scenarios is relevant to the common nature of uncollected or missing data in real world studies. Therefore, it is essential to establish relevant thresholds for common sources of bias in real world data and better specify the conditions which treatment effect conclusions may hold.”

Comment #5

Lastly I would prefer to see numbers at risk under the K-M plots where appropriate

Thank you for your suggestion. We have updated our K-M plots to include the numbers at risk under the respective plots.

Reviewer #2 (Remarks to the Author)

General comment: The solution to finding treatment effects in rare populations is ripe for new and creative methods. When found such methods can transform medicine dramatically for patients with rare diseases or rare disease subtypes. The authors have presented their strategy to approach this issue. The idea is one of great interest. This article attempts to present a strategy for identifying the treatment effect of pralsetinib, which is a highly selective RET inhibitor used in an open-label single-arm trial, compared to real world data for best available therapy. The manuscript itself has issues that need to be addressed before it accomplishes what the authors claim. If meaningful revisions can be completed, the strategy could be useful and interesting.

Strengths:

- 1) The authors have compiled meaningful clinical trial data and real-world data, have considered multiple sources of bias when combining real world data with current single arm trial data, and have included a reasonable series of methods to address them.
- 2) The topic is of great interest and timely.
- 3) The methods used are available in existing software so can be reproduced.
- 4) The conclusion is well thought and consistent with the findings.

We thank the reviewer for the insightful review and pointing out to the strengths of our paper.

Weaknesses:

Comment #1

1) The abstract, article, and study populations focus on identifying treatment effect of pralsetinib for the rare subtype RET fusion-positive aNSCLC using data from a current single arm trial and 3 real world data sets, 2 of which do not have RET fusion-positive information. These are assumed to be 99% RET fusion negative given the population estimate with an assumption that RET fusion status is not prognostic. A third dataset has RET fusion status. and is used to compare best available therapy. The statistical methods and results section of the manuscript focus almost entirely on the large datasets. In the results, additional information from the KEYNOTE-42 study are introduced without being included in the population description or methods. All results for the rare patients are relegated to the supplementary information, but this finding was the point of this investigation as described from the start. This leaves a disjointed story from start to finish, that is confusing to follow and not in line with the stated purpose of the manuscript.

Thank you for the feedback. We have added the results for the comparison involving the RET fusion-positive groups to the main article, and have adjusted the structure and language to better convey our intention that in addition to investigating the relative effectiveness of pralsetinib by comparing outcomes for RET fusion-positive patients receiving first-line (1L) pralsetinib in the ARROW trial with synthetic control arms (SCAs) derived from real-world data (RWD), equally important is our second goal. The second of the two goals of this study is to demonstrate how we quantitatively assessed the robustness of our findings to potential sources of bias in a comprehensive and systematic fashion to act as a guide for future SCA studies using RWD. This goal was motivated by call outs from regulatory and HTA bodies when submissions have involved RWD.

Comment #2

2) The methods described take advantage of existing methods, so this is not novel. This paper applies existing methods in a logical order that accomplishes the desired estimate of treatment effect.

Thank you for the feedback; indeed some of our methods were implemented based on existing work, and to our knowledge, the idea behind the QBA of hazard ratio robustness for poorer RWD performance is a novel way of considering whether discrepancies in RWD and key clinical trial performance may affect the conclusions drawn from the study. We have added language to better convey this and describe the motivation for the specific methods used:

“To our knowledge, this approach to answer how robust a treatment estimate is against any advantages conveyed to by poorer performance of treatments in real-world and trial settings has not been done previously. Such situations are often observed, with multiple cases in the context of immunology treatments in NSCLC alone.”³¹

Despite the wide range of possible QBA methods, many studies have yet to make QBA a default part of the analysis. Thus, overall we are looking to demonstrate the value of these methods for SCA analyses in the context of a study that was executed. The revised manuscript aims to better express this goal. Regulators and health technology assessment bodies have called for the evaluation of robustness through bias assessment, which has never been done before previously in SCA submissions.

Comment #3

3) These methods only work when inclusion in the rare subgroup is not prognostic.

Thank you for recognizing the potential impact of RET status, which we agree is an assumption made in the study. We will add language pertaining to RET+ status in our introductory sections to better motivate this decision.

Thank you very much for the detailed comments—

Comment #4 – Specific recommendations

- 1) If the focus is to be the treatment effect for the rare patient subtype, much of the supplemental material would ideally be moved to the main article and vice versa. Thank you for this comment and have added language explaining the focusses of the article to better motivate the material in the main article vs supplementary.
 - a. Supplemental Figures 1 and 2 could be combined into a single figure in the main article and replace most of the text from lines 84-123. This figure should also include the CGDB patients. Reading the text without the figure was challenging and the figure is much more accessible.
 - Thank you. We will consolidate the figure and add the CGDB patients
- 2) Supplementary material rows 246-277 appears to be the results that match the intention of the manuscript and would be better served in the article itself.
 - Thank you for the suggestion. We will be clarifying the intention of the manuscript more clearly as described in Comment #2, which would explain why these supplementary results were not included in the main article. Given the literature available, we believe that the limited prognostic value of RET is appropriate as an assumption for this study.
- 3) Article Figure 1 shows the effect of large datasets that do not represent the effect in the rare population. This is an important step in the process but could be in the supplemental materials.
 - Thank you for the suggestion. Similar as above, we will be clarifying the intention of the manuscript more clearly as described in Comment #2, which would explain why this figure was included in the main article. Regarding the intention of the manuscript and assumption regarding RET status, these were based on literature with evidence that RET fusion status was not prognostic, and for our equally important goal of demonstrating a complete example where the value of QBA methods for SCA analyses could be fully seen in the context of a study that was executed.
- 4) Article Figure 2 could be moved to supplemental materials.
 - Thank you for this suggestion. After making the changes described in Comment #2, we hope the motivation for leaving this figure as is in the main section is clearer since it represents an important analytical piece.
- 5) An additional supplemental figure could be meaningful if you could use the CGDB data for RET fusion negative patients to confirm general similarity for best available treatment for both negative and positive patients to confirm that your data support that RET fusion status is not prognostic.
 - We appreciate your suggestion for BAT similarity in RET-negative to implicate RET status as not prognostic. Prior literature using Flatiron CGDB data such as Hess et al. (2021) has provided evidence that RET is

not prognostic, which motivated the assumption in the study.

- 6) Small inconsistencies or omissions that can be easily addressed.
 - a. The number of acronyms makes reading difficult. I recommend keeping only a small number of universally familiar acronyms throughout.
Thank you for the suggestion, and we will focus our update on cutting down on acronyms.
 - b. Lines 10 and 42. aNSCLC is defined as “advanced NSCLC” in line 10 and as “advanced RET fusion-positive NSCLC” in line 42.
Thank you for catching this inconsistency; we have updated and removed the second reference to the acronym of aNSCLC.
 - c. Line 81. Please define WCG
Thank you for catching this. We have updated to WIRB-Copernicus Group.
 - d. Lines 239-240. With only 10 patients in the best available treatment RET fusion-positive cohort, I recommend not mentioning statistical significance.
Thank you for the suggestion, and indeed we are in full agreement. We updated that piece, and every other reference to significance and the BAT by updating to sample size referencing.

e. Line 292 – the KEYNOTE study shows up here for the first time. A little more justification and introduction for the reader is needed to justify its use, since it is not mentioned in any methods or patient populations above.

Thank you for the suggestion. We added more to the introduction and justification to the population section for the KEYNOTE trials:

“Digitised approximations of the Kaplan-Meier curves corresponding to two final phase-3 KEYNOTE trial arms (KEYNOTE-42 and KEYNOTE-189)^{21, 22} for pembrolizumab monotherapy and pembrolizumab with chemotherapy used for sensitivity analyses. Their purpose was to assess for each of the two regimens the impact of any discrepancy in overall survival between the corresponding clinical trial and RWD cohorts.”

f. Line 371 – While there are SMD <0.1 in the table, none are highlighted in green. This statement and table need to be consistent.

Thank you for recognizing this omission. We have updated any additional SMD<0.1 in green to be consistent.

g. Line 372. I recommend defining ESS in the table or as a note below the table. Also, I recommend including the actual N after trimming in addition to the ESS in this table. ESS is traditionally used for determination of power and does not prevent providing the actual N.

Thank you for catching this oversight. We have defined ESS below the table and appended the trimmed n’s (with an additional note) in the table.

h. Kaplan-Meier figures – I recommend adding at risk tables to all such figures. Thank you for your suggestion. We have updated our K-M plots to include the numbers at risk under the respective plots.

Reviewers' Comments:

Reviewer #1:

Remarks to the Author:

Thank you to the authors for making the changes to the manuscript in response to my and the other reviewers comments.

Overall I think it gives a much more balanced discussion of their methodology and findings.

I also found it much easier to read and understand the purpose of some of the statistical analysis performed.

I would be happy that this is now acceptable for publication and likely to be of interest to a wide range of readership

Reviewer #2:

Remarks to the Author:

General comment: The authors have responded in substantial and meaningful ways to both editors' comments. This now accomplishes a meaningful work in addressing the challenges of combining current and historical information for treatment effect in rare, non-prognostic patient subgroups in a consistent and clear story start to finish. All the strengths previously noted remain and the weaknesses abated. Some minor recommendations stand out in the revision.

Specific recommendations:

1) Since this method is only reasonable in the setting of a rare feature that is *not prognostic*, this needs to be explicitly clear to the reader from the start. I recommend this be added briefly but explicitly in 2 places - to the abstract when stating that the methods here can be used as a template and in the conclusion.

2) I recommend adding a paragraph in the discussion about the importance of RET+ being not prognostic for these methods to work and that other methods are still needed when the rare feature is prognostic.

3) In the response to reviewers, the authors state "Prior literature using Flatiron CGDB data such as Hess et al. (2021) has provided evidence that RET is not prognostic, which motivated the assumption in the study." This statement would be very useful to the readers. I recommend adding this statement of motivation to the manuscript in lines 106-109 while maintaining the caution of small samples sizes stated in Hess et al. (2021). Then point to additional references to add strength to the assumption.

4) Table 3 header indicates why some SMD are green, but Table 2 header does not. Table 2 header needs to include this as well.

5) Figure 1. For B, text is very small and several items stated longer or differently from similar ideas in A and C. Recommend making B more consistent with A and C in text size, box language, and format.

6) Appendix Figures 3 and 5 still need at risk numbers

REVIEWERS' COMMENTS

Reviewer #1 (Remarks to the Author):

Thank you to the authors for making the changes to the manuscript in response to my and the other reviewers comments.

Overall I think it gives a much more balanced discussion of their methodology and findings. I also found it much easier to read and understand the purpose of some of the statistical analysis performed.

I would be happy that this is now acceptable for publication and likely to be of interest to a wide range of readership

Thank you on behalf of the whole study team—we've greatly appreciate the comments and suggestions made.

Reviewer #2 (Remarks to the Author):

General comment: The authors have responded in substantial and meaningful ways to both editors' comments. This now accomplishes a meaningful work in addressing the challenges of combining current and historical information for treatment effect in rare, non-prognostic patient subgroups in a consistent and clear story start to finish. All the strengths previously noted remain and the weaknesses abated. Some minor recommendations stand out in the revision.

Thank you on behalf of the entire study team—we've greatly appreciate the feedback, particularly those regarding readability and how to better convey our message.

Specific recommendations:

1) Since this method is only reasonable in the setting of a rare feature that is *not prognostic*, this needs to be explicitly clear to the reader from the start. I recommend this be added briefly but explicitly in 2 places - to the abstract when stating that the methods here can be used as a template and in the conclusion.

Thank you for the suggestion. For the specific application in the study, the assumption that that the prognostic value of RET is limited was made as one of our datasets did not have RET status information. The methodology used does not depend on the rare feature in question not being prognostic, as this can be accounted for when selecting the study population as part of the eligibility criteria. We discuss the reason why we need this assumption in the discussion and hope this clarification sheds further light on the matter.

2) I recommend adding a paragraph in the discussion about the importance of RET+ being not prognostic for these methods to work and that other methods are still needed when the rare feature is prognostic.

Thank you for the feedback, please see the response above.

3) In the response to reviewers, the authors state “Prior literature using Flatiron CGDB data such as Hess et al. (2021) has provided evidence that RET is not prognostic, which motivated the assumption in the study.” This statement would be very useful to the readers. I recommend adding this statement of motivation to the manuscript in lines 106-109 while maintaining the caution of small samples sizes stated in Hess et al. (2021). Then point to additional references to add strength to the assumption.

Thank you for the suggestion. We have this reference as well as additional references in the introduction, study population, and discussion.

4) Table 3 header indicates why some SMD are green, but Table 2 header does not. Table 2 header needs to include this as well.

Thank you for the catch. We have made the tables and their headers consistent and black and white.

5) Figure 1. For B, text is very small and several items stated longer or differently from similar ideas in A and C. Recommend making B more consistent with A and C in text size, box language, and format.

Thank you for the feedback. We have simplified some of the boxes, though the differences between B and C are due to the differences in their eligibility criteria (due to differences in the datasets) and so will not be consistent.

6) Appendix Figures 3 and 5 still need at risk numbers

Thank you for the reminder. We have added the at risk tables to supplementary figures 3 and 5